# Capturing dense shelf water cascading with a high-resolution ocean reanalysis

Helena Fos[1], Jesús Peña-Izquierdo[2], David Amblas[1], Marta Arjona-Camas[1,3], Laia Romero[2], Víctor Estella-Pérez[2], Cristian Florindo-Lopez[2], Antoni Calafat[1], Marc Cerdà-Domènech[1], Pere Puig[4], Xavier Durrieu de Madron[3] and Anna Sanchez-Vidal[1]

[1]GRC Geociències Marines, Departament de Dinàmica de la Terra i de l'Oceà, Universitat de Barcelona, Barcelona, Spain
[2]Lobelia Earth SL, Barcelona, Spain.
[3]CEFREM, UMR-5110 CNRS-Université de Perpignan Via Domitia, Perpignan, France.
[4]Institut de Ciències del Mar, CSIC, Barcelona, Spain.

*Correspondence to*: Helena Fos (hfos@ub.edu)

**Abstract.** Dense shelf water cascading (DSWC) is an oceanographic process occurring when dense shelf water overflows the shelf edge downslope towards the deep sea. Monitored in the Northwestern Mediterranean by moorings since 1993 in the Lacaze-Duthiers Canyon and since 2005 in the Cap de Creus Canyon, numerical modeling with reanalysis extends this timeline further into the past. This study investigates a regional reanalysis (1987 – 2021) validated against mooring observations at 750-1000 m depth. The reanalysis successfully reproduces observed Intense DSWC (IDSWC) events from 1999, 2000, 2005, 2006, 2012, 2013, and 2018, while identifying one previously unreported event in 1987, and detecting no IDSWC between 1988 and 1998. The reanalysis effectively matches 84% of observed IDSWC days within the same week and 56% on the exact date. Instead of assimilating IDSWC events from mooring observations to resolve the cascading process, the model relies solely on the seawater density on the shelf and revealed the seawater properties along the canyon that caused IDSWC. This work highlights the importance of high-resolution reanalyses to investigate mesoscale processes impacting on larger scales in the deep ocean.

## 1 Introduction

Dense Shelf Water Cascading (DSWC) is a localized oceanographic phenomenon in selected regions worldwide, characterized by the spilling and sinking of shelf water due to its increased density caused by cooling, evaporation, or freezing (Ivanov et al., 2004). Since these overflows are driven by air-sea interactions, sensitive to climate change, in-situ monitoring programs are key to detect alterations impacting deep water ventilation and threatening related deep-sea ecosystems. The broad shelf of the Gulf of Lion (GoL) in the Northwestern Mediterranean Sea (Fig. 1a), is a well-known site for dense coastal water formation during cold winters caused by heat loss of surface water through the blowing of the dry and cold northerly winds, namely Tramuntana and Mistral (Millot, 1990). This dense shelf water flows cyclonically southward to the Cap de Creus peninsula. Then, it is forced to turn, overflow the shelf edge, and sink downcanyon if dense enough, channelized through submarine

canyons: mainly the Cap de Creus Canyon (CCC) and Lacaze-Duthiers Canyon (LDC) (Dufau-Julliand et al., 2004; Ulses et al., 2008a). The intensity of these cascades varies annually, and hence the maximum depth reached, from a few hundred meters (Arjona-Camas et al., 2025) to over 2000 m (Durrieu de Madron et al., 2023). During Intense DSWC (IDSWC) events, the shelf-water potential density ($\rho_\theta$; $\sigma_\theta = \rho_\theta - 1000$ kg/m$^3$) exceeds that of the Eastern Intermediate Water (EIW; Schroeder et
al., 2024) ($\sigma_\theta$: 29.05 - 29.10 kg/m$^3$) found between 300 and 800 m depth, which in previous studies was referred to as Levantine Intermediate Water (LIW), and even that of the Western Mediterranean Deep Water (WMDW; MEDOC Group, 1970) ($\sigma_\theta \sim$ 29.10 - 29.16 kg/m$^3$) found deeper, enabling the cascading to the bottom of the basin floor, over 2400 m depth. IDSWC in GoL was observed in 1999, 2005, 2006, 2012 and 2013 with mooring lines (Béthoux et al., 2002; Canals et al., 2006; Durrieu De Madron et al., 2013; Heussner et al., 2006; Palanques et al., 2006), hydrographic profiles (Puig et al., 2013), and gliders
(Durrieu De Madron et al., 2013; Testor et al., 2018). Although these observations enable studying specific DSWC characteristics, a broader climatological perspective is needed, especially as climate projections under the IPCC A2 scenario predict a 50-90% decline in DSWC in the GoL by the end of the century (Herrmann et al., 2008).

This work aims to study, for the first time, the climatology of IDSWC using multidecadal observations together with an ocean reanalysis, which combines model simulations of past hydrography and dynamics with observational data assimilation. Since
IDSWC observations are sparse and not assimilated by the reanalysis, they can be used to validate the underlying model resolving IDSWC. This approach provides decades of continuous daily data, revealing IDSWC in pre-observational years. Reproducing the IDSWC is challenging because reanalysis products typically exhibit reduced vertical resolution in the deep sea, hindering the reproduction of seafloor morphology and dynamics of submarine canyons where IDSWC occurs. This study is focused on the modeling of the CCC and LDC hydrodynamics during IDSWC, holding about two decades of in-situ
observational data.

## 2 Data and Methods

This study uses the Mediterranean Sea Physics Reanalysis (hereafter MedSea) (Escudier et al., 2020, 2021) provided by the E.U. Copernicus Marine Environment Monitoring Service (CMEMS). This product is generated by a hydrodynamical model based on the Nucleus for European Modelling of the Ocean (NEMO) version 3.6 (Madec et al., 2016), and corrected by
variational ocean data assimilation of in-situ salinity and temperature observations and sea level anomalies from satellite observations. According to the CORA dataset (Szekely et al., 2024) and SeaDataNet (https://www.seadatanet.org/) observational database, from which this reanalysis assimilates their salinity and temperature, the reanalysis does not assimilate any IDSWC observations. The model is initialized with climatological values from the C-GLORSv5 global ocean reanalysis (Storto and Masina, 2016). Boundary conditions are provided by the ERA5 reanalysis in the atmosphere (Hersbach et al.,
2020), by the C-GLORSv5 in the Atlantic, and by monthly mean climatological datasets for river runoff (more details in Escudier et al., 2021). The model bathymetry is started by the GEBCO 30 arc-second grid (Weatherall et al., 2015). Due to the high horizontal resolution of 1/24º (~4 km), the reanalysis enables the representation of submarine canyons and inner shelf

waters, with 141 vertical levels from 2 m width at the surface to 35 m at 1000 m depth. It provides daily averages from 1987, and this study focuses on the 1987-2021 period.

To validate the reanalysis results, we used in-situ observations from current meters, with temperature and current speed sensors moored 30-20 m above the seabed, at 750 and 1000 m depth in CCC, and at 1000 m in LDC (hereafter CC750, CC1000 and LD1000; Fig. 1). . The observations spanned 1993-2021 for LD1000, winter 2006 and 2011-2021 for CC1000, and winter 2005 for CC750. More details are shown in Table 1.

**Table 1**. Information about the datasets used in this study.

| | Reanalysis | In-situ observations (moorings) | | |
|---|---|---|---|---|
| Abbreviation | MedSea | LD1000 | CC1000 | CC750 |
| Zone (canyons) | Lacaze-Duthiers & Cap de Creus | Lacaze-Duthiers | Cap de Creus | Cap de Creus |
| Period | 1987-2021 | 1993 - 2021 | 2006; 2011-2021 | 2005 |
| Temporal resolution | Daily averages | 60 min | 30, 20 and 15 min | 20 min |
| Depth | Surface to bottom | 1000 m | 1000 m | 750 m |
| Distance from seafloor | - | 20-30 m | 20 – 30 m | 20 – 30 m |
| Variables | Speed, $\theta$, S, ($\sigma_\theta$ from S and $\theta$) | Speed, $\theta$ | Speed, $\theta$ | Speed, $\theta$ |


The bottom canyon locations at 1000 m depth in reanalysis data are 10 km downstream of the moorings, because the model resolution cannot resolve the sharp canyon incisions where the moorings were deployed (Fig. 1b). However, these different locations were chosen to capture the strongest DSWC events with signatures beyond the EIW layer. There, cascading shelf water was detected and defined with daily mean potential temperature ($\theta$) $\leq 12.6°C$ (after detrending) and speed $\geq 0.1$ m/s, for

reanalysis and observations. For reanalysis-derived IDSWC transport through the sections in Figure 1a, additionally, salinity (S) $\leq 38.44$ (fresher than WMDW), and $\sigma_\theta \geq 29.05$ kg/m$^3$ (denser than EIW) were chosen as dense shelf-water characteristics.

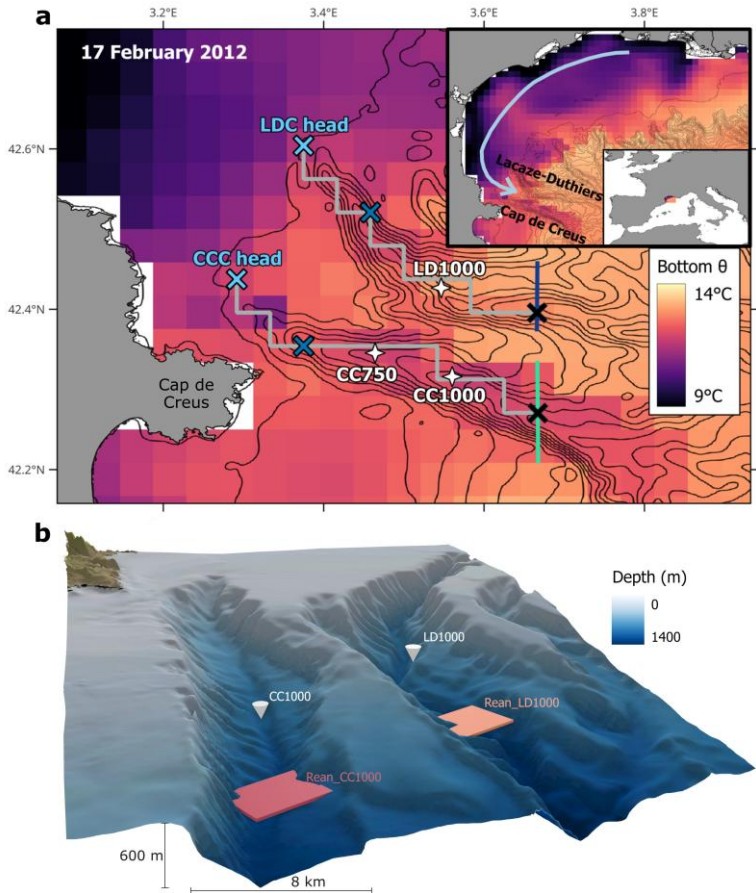

**Figure 1. (a) Map with the mooring locations (white crosses) and near-bottom temperature from the MedSea reanalysis on 17-02-2012 (see Movie S1 and S2). Blue and turquoise N-S-oriented lines indicate the cross-sections used for IDSWC transport calculations. Grey lines indicate the canyon axes following the reanalysis bathymetry, with crosses at 100 (light-blue), 400 (dark-blue), and 1000 (black) m bottom depth. (b) Scheme of the canyons with real bathymetry at 200 m resolution from EMODnet (https://emodnet.ec.europa.eu/) showing the mooring observation sites (white cones) and reanalysis data grid points (colored boxes), comparing the resolution of the reanalysis with the width of the canyons For clarity, the vertical axis is exaggerated by a factor of 5.**

### 3 Results

The interannual variability of IDSWC in the LDC and CCC is shown in Fig. 2, presenting the full time series of reanalysis and observational data at 750-1000 m depth. High-speed values associated with drops in temperature during the winters of 1987, 1999, 2000, 2005, 2006, 2012, 2013, and 2018 indicate the cascading of cold dense shelf water (Fig. 2a-d). Notably, no IDSWC events are found from 1988 to 1998. Overall, the reanalysis identifies the same IDSWC years as the observations and additionally captures an event in 1987. Note that in 2019, CC1000 observations showed a one-day-long anomaly, not considered as an overflow for this analysis.

IDSWC events occur simultaneously in both canyons, with more intense and faster transport of colder shelf water in the CCC, as indicated by both reanalysis and observations (Fig. 2a-d). Furthermore, the observed potential temperatures of both canyons

are strongly correlated (r = 0.78). Events tend to occur in paired years, as inferred from Durrieu de Madron et al., 2023, where the second year is weaker and usually hardly detectable in the LDC. The observations reveal a significant negative correlation between potential temperature and speed in CC1000 (r = -0.63), weaker for the reanalysis (r = -0.40). Moreover, the reanalysis correlates better with the observations for potential temperature (r = 0.41 in LDC, r = 0.61 in CCC) than for velocity (r = 0.21 in LDC, r = 0.29 for CCC), all significantly at the 95% confidence level. A detailed examination of cascading events is shown in Fig. 3.

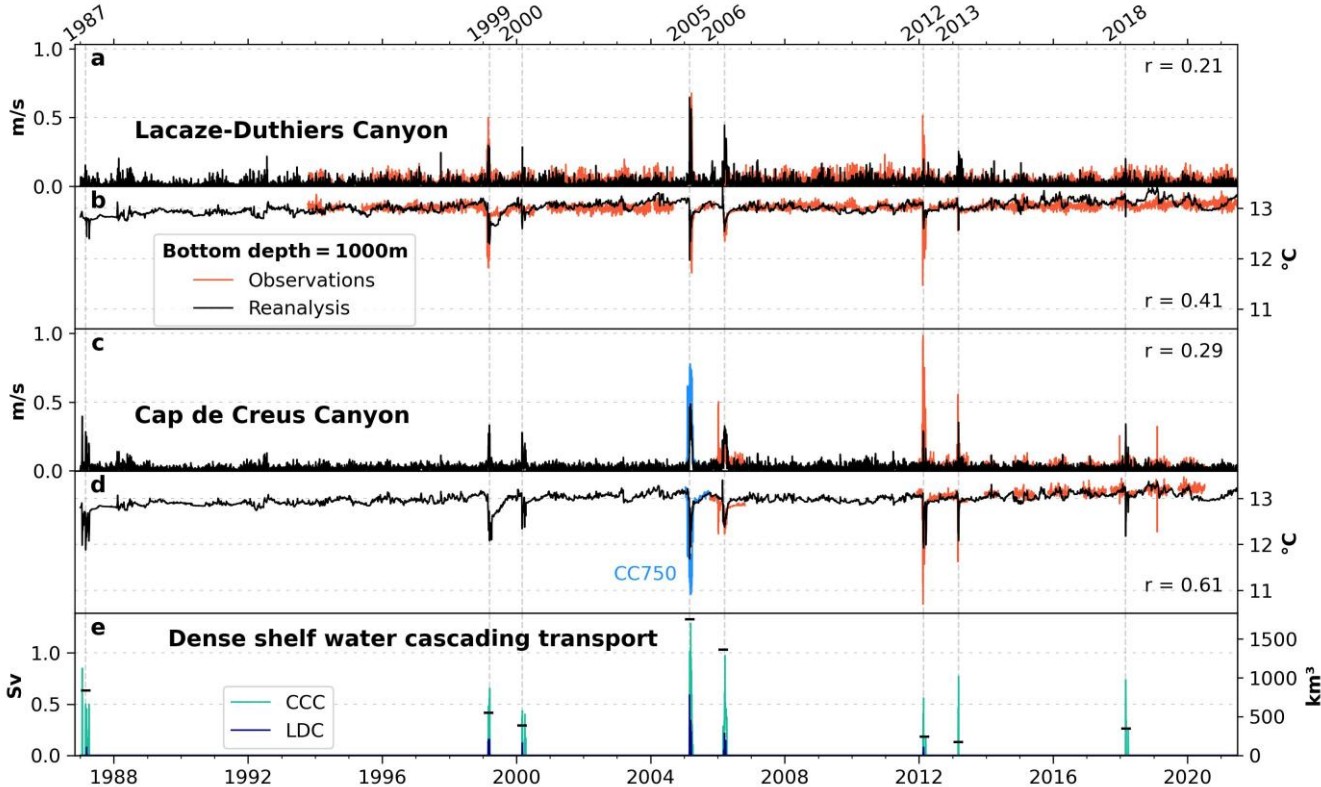

**Figure 2. Daily-mean (1987-2021) bottom current speed (a, c), and potential temperature (b, d) at 1000 m depth in the Lacaze-Duthiers Canyon (LDC; a-b) and Cap de Creus Canyon (CCC; c-d). 750-meter-depth observations (CC750) are shown in blue. Vertical grey dashed lines indicate the dates of annual potential temperature minima at CCC, indicating intense dense shelf water cascading (IDSWC) events, defined with $\theta \leq 12.6°C$ (after detrending) and speed $\geq 0.1$ m/s. In 2019, the observed anomaly in panels c and d lasted just one day, due to its short duration, it is not considered an IDSWC event in this study. Pearson correlation coefficients (r) between observations and reanalysis data. (e) Daily-mean IDSWC transport (Sv; 1 Sv = $10^6$ m³/s) at 1000 m for both canyons (CCC in turquoise, LDC in blue) from reanalysis data, computed for water with salinity $\leq 38.44$ and $\sigma_\theta \geq 29.05$ kg/m³. Total winter transport volume (km³) across both canyons is shown with black hyphen-shaped markers. The acronyms IDSWC, LDC, and CCC are used throughout the figures.**

Winters where the reanalysis standard deviations (SD) exceed the observations SD (Fig. 3i-j), mean an overestimation of the IDSWC and vice-versa. The higher the correlation, the better the timing agreement in IDSWC. Considering these two factors, the standardized SD error (SDE) (Wadoux et al., 2022) quantifies the discrepancy between the reanalysis and the observed IDSWC. In 2005 and 2012, the two strongest IDSWC events, the cascading lasted from late January/February to late

May/April, with measured instantaneous speeds exceeding 0.8 and 1.2 m/s, respectively. In these two years, the reanalysis detects temperature and velocity anomalies with high temporal correlation to observations but underestimates their intensities (standardized SD << 1). IDSWC replicas in 2006 and 2013 are also well reproduced concerning time and intensity, particularly for the CCC, with slightly lower accuracy for the LDC (Fig. 3). These prominent DSWC events of 1999, 2005, 2006, 2012, and 2013 are the best-reproduced (SDE: 0.7 - 1.5). However, in 2018, the reanalysis overestimates the duration of IDSWC (Fig. 3i), since the observations show only a short sub-daily anomaly, preceded by another short event in December (Fig. 2). Overall, the reanalysis IDSWC flow is 0.13 ± 0.05°C warmer and 30% weaker than the observations in the CCC, and 0.37 ± 0.15°C warmer but 6% stronger in the LDC.

Using the IDSWC definition criterion from section 2, between 2005 and 2020, 112 IDSWC days were observed at CC750 and CC1000, while the reanalysis identified 97 days (13% fewer). At LD1000, between 1993 and 2021, 50 IDSWC days were observed, compared to 39 in the reanalysis (22% fewer). The reanalysis performed better in the CCC (63% precision, 54% recall) than in the LDC (38% precision, 30% recall), meaning that over 60% and almost 40% of the simulated IDSWC days at CCC and LDC, respectively, could have actually occurred (Precision=Well_Predicted/Total_Predicted; Recall = Well_Predicted/Observed). Allowing a 7-day margin to stay in the same synoptic conditions improves the results: 78% of observed days with IDSWC are reproduced by the reanalysis in the CCC (84% in the LDC).

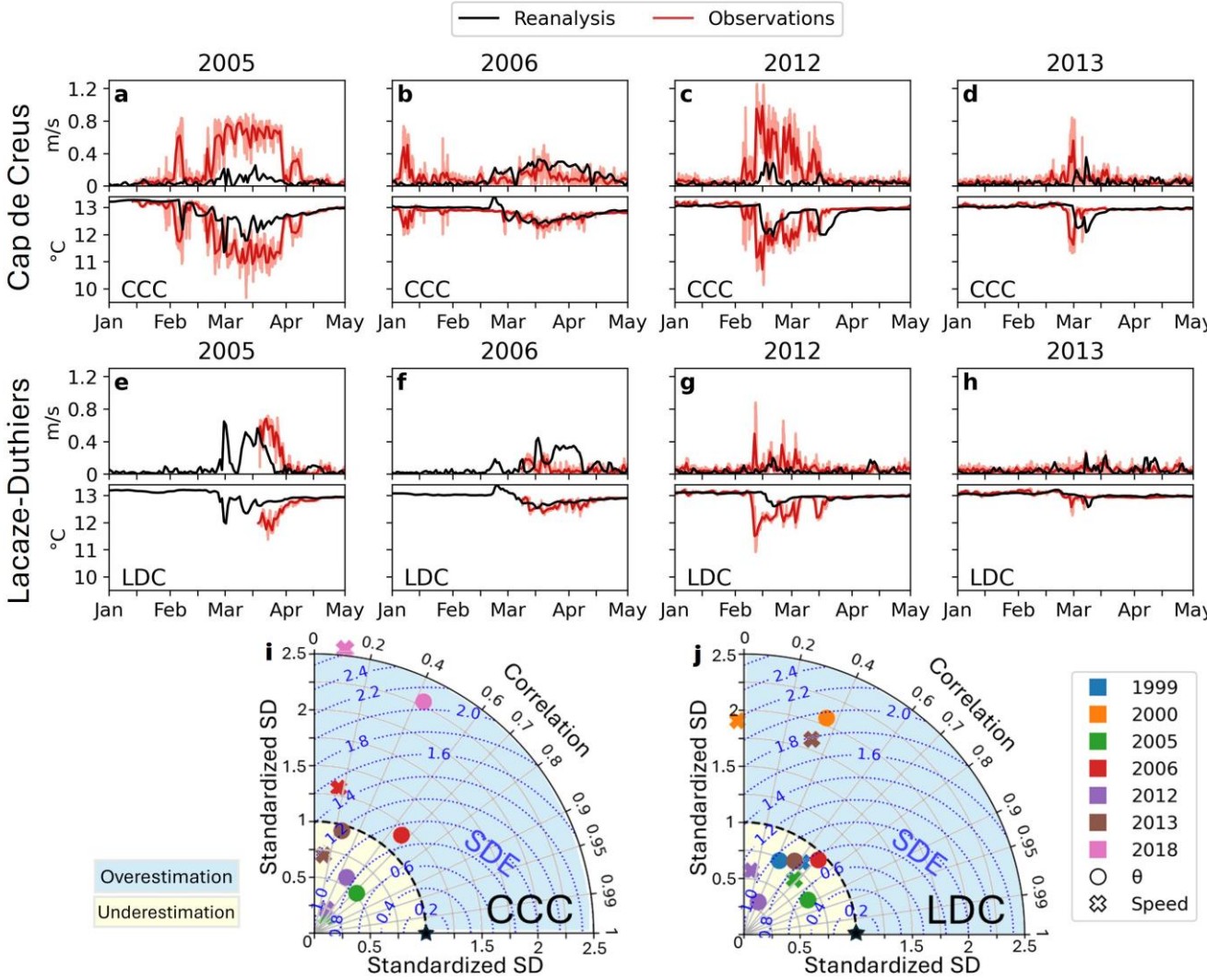

**Figure 3. (a–h) Comparison of daily IDSWC between observations and reanalysis. Sub-hourly measurements (light red), and their daily mean (bright red). The 2005 data corresponds to mooring CC750 and the reanalysis at 750 m in the CCC. (i–j) Taylor diagrams for data from panels a-h and other IDSWC events, where the reanalysis standard deviations (SD) are standardized to the SD of the observations (black star), and the standardized SD error (SDE) is obtained. All the correlations are significant (p < 0.05) except for the speed in 2000 and 2012 in LDC, and in 2006, 2013, and 2018 in CCC. Values for 2018 are not shown in panel j because the IDSWC event was not found in LDC.**

Peak DSWC transport values of 1.29 Sv are found in CCC, and 0.59 Sv in LDC for 2005 (Fig. 2e). The winter-integrated transport is also highest in 2005: 1750 km³ summing the two canyons. From 1987 to 2021, the IDSWC transport decreased by 59% and the DSWC days in CCC dropped by 60%, but Mann-Kendall tests show non-statistical significance at 90% confidence level.

Figure 4 shows how the cold ($\theta$ < 12.6ºC) and dense ($\sigma_\theta$ > 29.1 kg/m³) shelf water flows along the seafloor from the shelf edge

into both canyons. During a winter with IDSWC (Movie S3), the relatively fresh shelf water becomes colder until denser than

intermediate water and sinks until reaching neutral density. Firstly, shelf water sinks into intermediate depths, as seen in Fig.

4c and 4d where temperature and salinity values at 100 m depth match the values at 400 m depth. Then, shelf water continues

to cool and densify until reaching the density of the deeper water, and cascades more intensely. Consequently, the velocity at

1000 m increases and the temperature and salinity decrease. Peaks in density, speed and salinity, along with drops in

temperature at 100, 400 and 1000 m depth occur almost simultaneously, meaning cascading shelf water reached the deep basin

in less than 24 h.

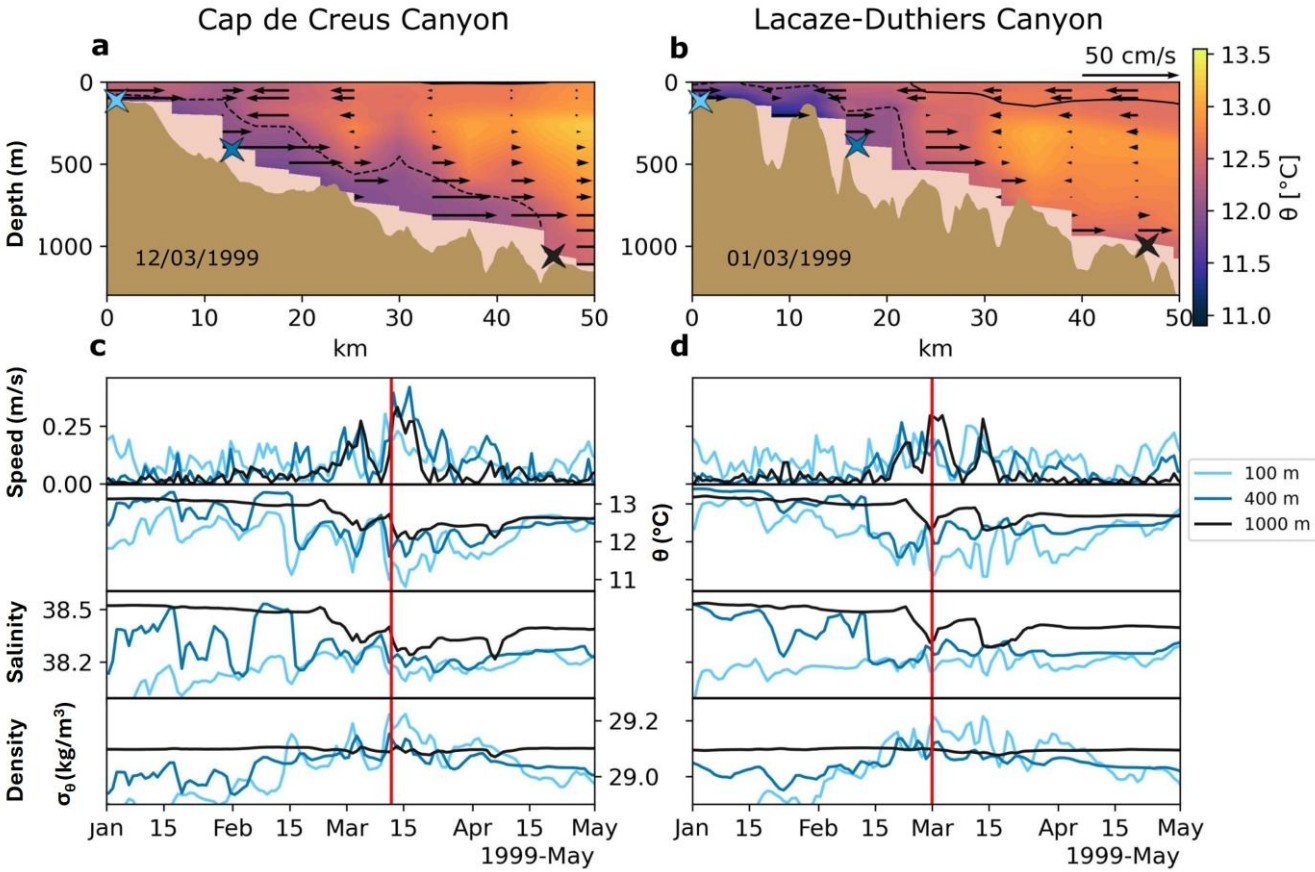

**Figure 4. (a-b) IDSWC in 1999 in CCC and LDC from reanalysis. The black dashed line marks the 29.1 kg/m³ isopycnal, and the solid black line, the 29.0 kg/m³ isopycnal. Black vectors show downcanyon flow. Blue crosses mark the canyon bottom locations**
**shown in c and d. The bathymetry corresponds to the reanalysis section represented with a grey line in Figure 1. (c-d) Time series of current speed, potential temperature, salinity and potential density. The red vertical line marks the date shown in a and b.**

Figure 5 shows shelf water exceeding $\sigma_\theta$ > 29.16 kg/m³ in 1987, 1999, 2000, 2005, 2006, 2012, 2013 and 2018, being denser

at the canyon heads than at the basin bottom. Therefore, the model resolves the IDSWC and propagates this dense water

downcanyon. In other years such as 1992, 2003 and 2010, the densest shelf water is colder than in 2000 or 2006, but fresher, making it less dense and preventing cascading.

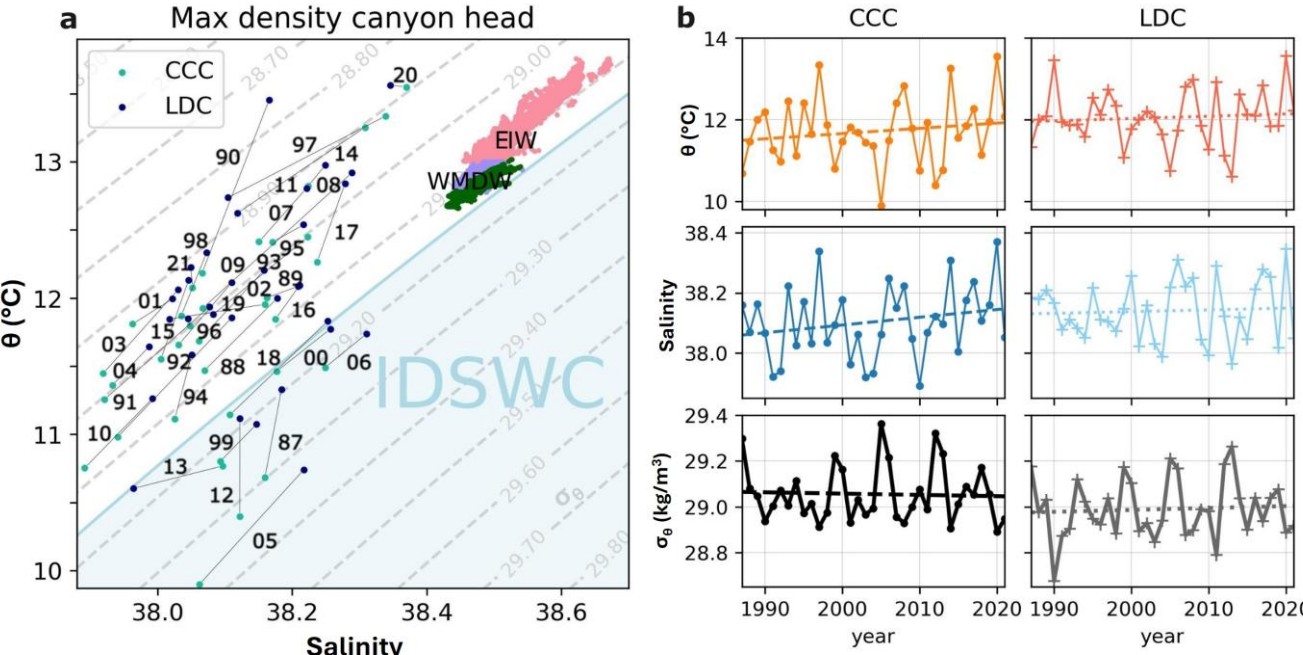

**Figure 5. (a) θ-S diagrams with reanalysis data for the day with maximum density annually at the canyon heads (100 m depth). Two-digit numbers indicate years, and the grey line connects canyon values. Eastern Intermediate Water (EIW) and Western Mediterranean Deep Water (WMDW) (pink and purple, respectively) are defined based on Juza et al. (2013) and correspond to the hydrographic properties at the bottom of CCC between 400 and 1300 m of depth, and WMDW at 2300 m in the basin (dark green). Blue shading highlights years with IDSWC, when canyon head potential density exceeds the maximum density of WMDW in our data ($\sigma_\theta > 29.16$ kg/m$^3$). (b) Annual maximum shelf-water density (bottom), salinity (middle), and temperature (top), in both canyons, with dashed lines showing trends (non-statistically significant).**

The maximum annual shelf-water potential density at the canyon heads (Fig 5a) shows that water in the CCC is generally colder and fresher, but shelf-water densities in CCC and LDC are typically almost identical, with CCC often slightly denser, or significantly as in 1987 and 2012. Both canyons exhibit densities sufficient to cascade deeply in 1987, 1999, 2000, 2005, and 2012, 2013, and 2018. Comparing these shelf-water densities over time (Fig 5b), no significant trend is found: the slight increase in temperature of 0.38ºC/30 years and 0.17ºC/30 years is compensated by an increase in salinity of 0.08/30 years and 0.02/30 years in the CCC and LDC, respectively.

## 4 Discussion

The MedSea reanalysis correctly reproduces IDSWC events, even with 84% recall for the correct week, due to the contribution of ERA5 atmospheric forcing, lateral advection, and assimilated shelf-water observations. Therefore, the NEMO model

succeeds in simulating opportune IDSWC given the right atmospheric and oceanic forcing conditions. This provides reliability
to the simulated IDSWC events in the pre-observational period. With its ~4 km horizontal resolution, the reanalysis identifies
submarine canyons (Fig. 1b) and simulates IDSWC. However, at 1000 m depth in the canyons, especially during intense events
such as in 2005 and 2012, the reanalysis temperature and speed are less extreme than the observed. This is mainly due to
resolution limitations, smoothing and oversimplifying the lateral and vertical hydrographic complexity of IDSWC (Fig. 1b),
and leaving the short-lived IDSWC flows poorly captured. The reanalysis aligns better with the observed IDSWC days in the
CCC, where the events are stronger, since the densest shelf water is accumulated by the Cap de Creus peninsula (Fig. 1, Movie
S1 and S2). Fig. 5a aligns with Fig. 2, 3 and 4, where colder IDSWC is found in CCC, but both canyons exhibit densities
sufficient to cascade in 1987, 1999, 2000, 2005, and 2012, 2013, and 2018. Both shelf-water temperature and salinity are key
in controlling shelf-water density and cascading (Fig. 5a). However, the reanalysis ignores the interannual variability in river
discharges, which may lead to possible discrepancies in salinity. The uncertainty in transport estimates arises from the lack of
full water-column observations, but IDSWC transport may be underestimated in years with fewer IDSWC days and lower
mean velocities than observed (e.g., 2012), and overestimated in the opposite case (e.g., 2018). The slight decline in the IDSWC
transport aligns with the observed IDSWC days, although more data are required to obtain a significative trend.

The long-lasting IDSWC event revealed in 1987 aligns with Béthoux et al. (2002), who suggested an IDSWC event after
January 1987 and no events from 1988 to 1998, as confirmed by the reanalysis The two most significant observed events
occurred in 2005 and 2012 (Figs. 2 and 3), as reported by Canals et al. (2006), Palanques et al. (2006), Sanchez-Vidal et al.
(2008), Ulses et al. (2008b), Durrieu de Madron et al. (2013) and Puig et al. (2013). The strongest export of IDSWC occurred
in 2005, the transport volume of which was estimated by Canals et al. (2006) at 750 km$^3$, by Ulses et al. (2008b) at over 1200
km$^3$, and 1750 km$^3$ in the present study.

## 5 Conclusion

The MedSea reanalysis is suitable for studying the IDSWC interannual variability in the NW Mediterranean, particularly in
LDC and CCC, where IDSWC occurred in 1987, 1999, 2000, 2005, 2006, 2012, 2013, and briefly in 2018 and 2019. The
reanalysis effectively captures IDSWC timing and intensity, revealing pre-observational events and transport variability in
both canyons. IDSWC intensity is higher in the CCC, and events tend to recur the following winter. Due to this longer time
series of reanalysis data, a 6-7-year frequency of paired IDSWC events is detected, interrupted during the early and mid-1990s.
Possible trends in IDSWC transport, duration, or shelf-water properties have appeared non-statistically significant, although it
should continue to be monitored, as some decline has been perceived in recent years.

The success of using reanalysis to describe known IDSWC episodes provides a valuable tool for studying IDSWC or other
overflows in recent years, even in the absence of observations.. The potential of the NEMO model to resolve IDSWC under
the right forcing or preconditioning factors may enhance the prediction of these events.

## Data availability

MedSea reanalysis data are available on the CMEMS website (https://data.marine.copernicus.eu/product/MEDSEA_MULTIYEAR_PHY_006_004/services; Escudier et al., 2020). LD1000 dataset (Durrieu de Madron et al., 2024) is available at https://campagnes.flotteoceanographique.fr/series/62/ and https://www.seanoe.org/data/00349/45980/. CC1000 (Sanchez-Vidal et al., 2025) and CC750 (Puig & Palanques, 2025) datasets are available at https://www.seanoe.org/data/00936/104746/ and https://www.seanoe.org/data/00936/104799/ respectively.

## Video supplement

Movie S1. Bottom θ from the Mediterranean Sea Physics Reanalysis in winter 2005. White markers represent the locations of the moorings.

Movie S2. Same as Movie S1 but for winter 2012.

Movie S3. Daily sequence of the hydrographic properties in the vertical sections of Cap de Creus and Lacaze-Duthiers canyons, as in Fig. 4, but for all the winters with IDSWC.

These movies are all available at https://av.tib.eu/series/1951.

## Author contributions

The study was carried out under the supervision of JPI, ASV, DA and LR, with the assistance of VEP, CFL, MAC, XDM, and PP. Mooring data acquisition and processing were performed by AC, MCD, ASV, DA, XDM, and PP. HF conducted the analysis and wrote the original draft, with contributions from all co-authors in reviewing and editing.

## Competing interests

The authors declare that they have no conflict of interest.

## Acknowledgements

We acknowledge the support from the Spanish government through the grant FAR-DWO PID2020-114322RBI00 funded by MCIN/AEI/10.13039/501100011033; and the Generalitat de Catalunya through the Excellence Research Groups Grants 2021-SGR-01195 and 2021-SGR-00433, the Secretaria d'Universitats i Recerca from Departament de Recerca i Universitats for supporting the FI-SDUR fellowship 2023-FISDU-00233, and the Direcció General de Política Marítima i Pesca Sostenible in the framework of the collaborative agreement between Universitat de Barcelona and Departament d'Acció Climàtica, Alimentació i Agenda Rural. ASV acknowledges the support by Catalan Institution for Research and Advanced Studies

ICREA. This work acknowledges the Severo Ochoa Centre of Excellence accreditation (CEX2019-000928-S). We acknowledge the MOOSE program (Mediterranean Ocean Observing System for the Environment) funded by CNRS-INSU and the Research Infrastructure ILICO (CNRS-IFREMER). Sara Iftikhar is also acknowledged for providing the easy-mpl python package. We thank the two anonymous reviewers for their valuable comments. Finally, we would like to thank the
crew from the vessel Lluerna and all the people involved in the CC1000 and LD1000 mooring operations during almost 2 decades of efforts at sea to maintain these long-term deployments that are crucial to detect changes in the ocean.

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
