# Peer review of "Capturing dense shelf water cascading with a high-resolution ocean reanalysis"

_EGUsphere, 2025_

## Author Comment (AC1)

**Solving dense shelf water cascading with a high-resolution ocean reanalysis**

This document contains our responses to the two referees or reviewers. We want to thank both reviewers for all their valuable comments. We have addressed all their comments, starting with the reviewer 1, with their comments in black and our replies in blue, and then the same for the reviewer 2.

**Reviewer 1:**

**Review of «Solving dense shelf water cascading with a high-resolution ocean reanalysis» by Fos et al.**

The authors investigated the dense shelf water cascading (DSWC) process at two submarine canyons in the Gulf of Lion (northwestern Mediterranean Sea). The aim of the study is to evaluate the performance of the reanalysis in representing the DSWC. To achieve this goal, the authors compare the reanalysis data with in-situ observations from two moorings over the period 1993-2021 for Lacaze-Duthiers Canyon and 2011-2021 for Cape de Creus Canyon. From the results, they conclude that the MedSea reanalysis is able to successfully reproduce the interannual variability of the DSWC, although the timing and intensity of the phenomena are biased at the daily scale.

I found the topic interesting because the reanlysis performance in deep layers has not been thoroughly evaluated for the Mediterranean Sea. Thus, testing reanlysis skills in representing an oceanic proccess such as the DSWC is a good excersise. The manuscript is generally well written and the results are relevant to the scientific community, especially the Mediterranean community. However, I found some points that should be improved before its publication in OS.

1) Abstract : I find the statement "...can be used to find other unreported cascading events elsewhere" somewhat problematic. While the approach is compelling, it's important to note that the reanalysis product used in this study is specifically designed for the Mediterranean. This does not necessarily imply that other reanalysis products (e.g. global) would perform equally well in capturing cascading processes in different regions. In addition, it may be premature to extend the conclusions to other cascading areas within the Mediterranean basin, such as the Adriatic or Aegean Seas, given the different water mass characteristics and potentially different local dynamics in these regions. It would be helpful to evaluate reanalysis performance in these areas before making broader generalizations. Please rephrase.

Thank you for your comment, we find it very appropriate. We agree that we cannot know if this reanalysis could be successfully used to study the DSWC in the Aegean and Adriatic Seas. Because the reanalysis products have assimilated observations, they tend not to differ much from reality, and we believe that the resolution of the reanalysis is a key factor, as it needs to be adequate to resolve the spatial characteristics of the DSWC process. In our case, we used the Mediterranean Sea Physics Reanalysis, which is a regional reanalysis, because of its relatively high resolution of 1/24⁰ or 4 km approximately and 141 vertical levels. This resolution permits the dense shelf water formation in the Gulf of Lions and the reproduction of the IDSWC within these studied canyons. Other reanalysis products with a lower resolution, global reanalyses, for example, would likely perform worse in our area. However, these might be appropriate in other areas where the cascading process takes place on a larger spatial scale, for example, in larger shelves and slopes without canyons. Indeed, what we wanted to refer to is that a reanalysis product can be useful and helpful to study dense shelf water cascading in other areas, even though the different reanalysis products should be tested before, to evaluate their performance. To avoid confusion, we have deleted this part of the abstract where we generalize the use of reanalysis to other areas, and we have added a sentence emphasizing the importance of the resolution. Find the last 2 lines of the abstract with the modification.

2) Introduction: Since the manuscript was previously submitted to another journal with strict space limitations, I understand why the introduction may have been kept concise. However, Ocean Science does not have such constraints, and I found the current introduction somewhat brief and missing some important background information. I would recommend that the authors provide a more detailed description of the state-of-the-art, including the criteria (e.g., temperature, salinity, potential density, current velocity) used in previous studies to characterize DWSC in the Gulf of Lions, as well as a more thorough geographical description of the study area. I also strongly recommend including references when referring to water mass characteristics (e.g., LIW, WMDW) and their typical depths in the water column. I suggest more clearly highlighting the novelty of this work in comparison to previous studies. Finally, a short paragraph at the end of the section introducing the content of the next section might also be nice.

Thank you for your comments. You are right, a longer and more detailed description in the introduction could offer additional context, however we have intentionally kept it concise to maintain the focus on the key elements of the study.

Regarding the water masses, we have now added references. We have replaced the LIW for EIW, which stands for Eastern Intermediate Water (Schroeder et al. 2024). This water we find in the Western Mediterranean is not just LIW but a mixture with

Cretan Intermediate Water (CIW) according to Schroeder et al. (2024), which suggests updating the name to EIW instead of LIW, used in many previous studies. We have added this reference in line 35 of the marked manuscript. And for WMDW, we have included the reference to MEDOC Group, 1970, in line 36.

The full references have been added to the reference list:

MEDOC Group: Observation of formation of deep water in the Mediterranean Sea, 1969, Nature, 227, 1037-1040. https://doi.org/10.1038/2271037a0, 1970.

Schroeder, K., Ismail, S. Ben, Bensi, M., Bosse, A., Chiggiato, J., Civitarese, G., Falcieri, F. M., Fusco, G., Gačić, M., Gertman, I., Kubin, E., Malanotte-Rizzoli, P., Martellucci, R., Menna, M., Ozer, T., Taupier-Letage, I., Yáñez, M. V., Velaoras, D., and Vilibić, I.: A consensus-based, revised and comprehensive catalogue for Mediterranean water masses acronyms, Mediterr Mar Sci, 25, 783–791, https://doi.org/10.12681/MMS.38736, 2024.

As for the novelty, we now have changed the first sentence in the last paragraph of the Introduction section to emphasize the novelty of this study: *"This work aims to study, for the first time, the climatology of IDSWC using multidecadal observations together with an ocean reanalysis, which combines model simulations of past hydrography and dynamics with observational data assimilation"*. Also, in the last sentence of the previous paragraph, we try to highlight the importance of this longer-term study compared to previous studies, which have focused on just one specific IDSWC event.

3) Data and Methods: I suggest to include a small table summarizing the main information related to the reanalysis data and those related to the measurements from in situ observations. This would help the reader to easily follow the methods section.

We agree that this addition will help guide the reader through the Methods section. We have added the Table 1, which summarizes the most important information about the datasets used in this study.

L68-71: It is not clear to me whether the criteria used to define DSWC events come from observations or from reanalysis. This should be clarified. Are you defining DSWC events from in-situ observations or are you moving to values (model reality) where reanalysis successfully reproduces DSWC?

Thank you for your comment. The same temperature and speed criterion is used for reanalysis and observations. It has been clarified in the text (line 77 of the marked manuscript).

4) Results: In general, I found this section to be very descriptive, I would recommend the authors to go a little bit more in depth in the analysis in order to provide answers to the open questions (such as, what mechanisms might be involved ?).

We agree that the analysis and interpretation of the underlying mechanisms to obtain these results are important. Our intention was to keep the Results section with a clear and objective description of the findings, and with the interpretation of the mechanisms involved in the Discussion section. For example, in the Discussion section, we explain that the reason why the 2005 and 2012 simulated events have a lower speed and higher temperature than the observations could be caused by the resolution of the model: "*This is mainly due to resolution limitations, smoothing and oversimplifying the lateral and vertical hydrographic complexity of IDSWC (Fig. 1b), and leaving the short-lived IDSWC flows poorly captured*".

- L84: Why is 2019 not considered a DSWC event? It seems clear from the observations that there was an event in the CCC in 2019. However, this event is not captured by the reanalysis.

You are right that the 2019 DSWC event was clearly captured, but it was too short; as it only lasted one day, we prefer not to consider it as an IDSWC event for our analyses. We have clarified this in the caption of Figure 2.

-L87: 'Events tend to occur in paired years, where the second year is weaker and usually hardly detectable in the LDC.'Any explanation for why DSWC occurs in paired years? Why is it more difficult to detect in LDC than in CCC?

Thank you for your comment. The fact that DSWC events tend to occur in paired years, with the second one usually being weaker, is a very interesting aspect. This pattern has also been noted by Durrieu de Madron et al. (2023), and now we have added this citation following the mention of these paired events. However, understanding the reasons behind it would require a detailed analysis of factors like preconditioning of shelf waters and atmospheric forcing over several years. Since our focus here is on validating the reanalysis product using long-term observations, as well as on inferring when IDSWC occurred in the past, we do not delve into this topic.

Regarding the second part of your question, we explain this in the Discussion section. Probably, the difference in how the cascading is more detected in the CCC in comparison with LDC is because the CCC is closer to Cap de Creus peninsula,

where denser shelf water is accumulated in greater volumes. The Cap de Creus peninsula acts as a topographic barrier that conducts the formed dense shelf water in the Gulf of Lion towards the slope in the south-western part of the gulf (see the grey arrow in Figure 1a), and mainly funnels down the Cap de Creus Canyon, but also the Lacaze-Duthiers Canyon. Due to the short word limit, we were unable to provide more details, but you can find more details in Ulses et al 2008a, which is cited in the text. Also, for a better understanding of this process, please watch the supplementary movies S1 and S2, which are also cited in the Discussion section (line 187 of the marked manuscript).

-Figure 2: Please use the same ranges on the y-axis to make LDC and CCC easy to compare. Avoid using acronyms in the caption (figure captions should be understandable to the reader without having to read the entire manuscript).

We have now adjusted the y-axes to be the same for LDC and CCC and be easier to compare both canyons.

Regarding the acronyms in the figure captions, you are right. We have now added the definitions of the acronyms IDSWC, CCC and LDC in the captions of Figure 2, where they are first used. Also, for clarity, we have added in the captions of Figure 2 the sentence: *"The acronyms IDSWC, LDC, and CCC are used throughout the figures",* making reference that these acronyms are defined here and we will not redefine them again in all figures, to avoid redundancy. In addition, CC750 has been defined in the captions of Figure 2; SD, in Figure 3; and the water masses LIW and WMDW, in Figure 5.

-Figure 3: Why is the year 2018 missing in 3j?

Since the IDSWC in 2018 only occurred in the Cap de Creus Canyon but not in Lacaze-Duthiers Canyon, and we want to see events happening in both canyons, we have considered not to include it in Figure 2. We have clarified this in the captions of Figure 3.

- L123: Here, after finishing the analysis of Figure 3, you come back to the analysis of Figure 2e, which seems a bit strange. Please reorganize. These volumes are calculated for waters $\sigma\theta \geq 29.05$ kg/m$^3$? Clarify in the caption of Figure 2.

You are right about the order. Nonetheless, we chose to follow this structure to maintain a logical flow in the narrative: we first compare observations and reanalysis (Figure 2a-d), then we compare the observed and simulated deep dense shelf water cascading events winter by winter (Figure 3), and then we analyse cascading days and potential trends. And afterwards, we also test the trend on cascading transport with reanalysis data, which cannot be directly compared

because we cannot calculate transport estimates with these observational data. Also, we describe the highest transport values that are in accordance to the strongest winters that we have seen in detail before. And this transport is in Figure 2 because it aligns well with the strongest and weakest IDSWC events by temperature and velocity of the whole time series at 1000m of depth.

Regarding the transport in the captions of Figure 2, we have now added the specification that the IDSWC are defined with $\theta \leq 12.6°C$ (after detrending) and speed $\geq 0.1$ m/s. And for the transport: "*(...) computed for water with salinity $\leq 38.44$ and $\sigma_\theta \geq 29.05$ kg/m³*".

5) Discussion: Have you identified any EMT signals in the reanlysis that prevent deep cascading from 1988 to 1998?

According to our results and the data we used, we cannot identify any EMT signals beyond the fact that, within the EMT period, no IDSWC would have occurred. According to Josey et al 2023, during the EMT period, while in the Eastern Mediterranean the dense water formation was enhanced, in the northwestern Mediterranean, the net heat flux was abnormally positive, meaning a decreased surface heat loss to the atmosphere. Additionally, beyond that period, it is plausible that the EMT event in the eastern Mediterranean led to a shift in water mass properties and circulation that, after a delay, significantly influenced the intermediate waters in the northwestern Mediterraneanan, and thus the intensity of dense water formation by open-ocean convection in 2005 and 2006, according to Schröder et al. (2006), and Li and Tanhua (2020). However, to what extent these two factors in the EMT would have influenced the depth of the DSWC is something still under investigation, and we cannot make any conclusive statement about this in the present study. In this paper, we want to reveal the past IDSWC by using a reanalysis, but the reason why these events did occur or did not, requires a more extensive study, which is also in progress, but it is out of the scope of the present paper.

To identify the EMT signals that prevented the IDSWC, we would need to explore the air-sea interaction and preconditioning for IDSWC. The purpose of this study is to provide a long temporal overview of IDSWC events thanks to the use of mooring data and a reanalysis. Reanalysis products are an valuable tool for representing variability and long-term trends in the deep sea, and with this study, we demonstrate that a publicly available CMEMS product can be useful to study dense water overflows. To avoid confusion, we have deleted the sentence where we refer to EMT in the Discussion and in the Conclusion sections.

References:

Josey, S. A., Somot, S., and Tsimplis, M.: Impacts of atmospheric modes of variability on Mediterranean Sea surface heat exchange, J Geophys Res Oceans, 116, https://doi.org/10.1029/2010JC006685, 2011.

Li, P. and Tanhua, T.: Recent Changes in Deep Ventilation of the Mediterranean Sea; Evidence From Long-Term Transient Tracer Observations, Front Mar Sci, 7, https://doi.org/10.3389/fmars.2020.00594, 2020.

Schröder, K., Gasparini, G. P., Tangherlini, M., and Astraldi, M.: Deep and intermediate water in the western Mediterranean under the influence of the Eastern Mediterranean Transient, Geophys Res Lett, 33, https://doi.org/10.1029/2006GL027121, 2006.

6) Conclusions : As I noted in the abstract, I would caution against generalization of conclusions without detailed analysis in other regions. In this section I would suggest to highlight the biased performance of reanalysis at the daily scale and the main deficiencies found. In this way you can provide some perspectives to suggest improvements to the reanalysis developers.

We have removed the phrase "and could be applied elsewhere" from the last paragraph to avoid overgeneralization. Regarding the bias in the reanalysis, in the Conclusion section we prefer to avoid generalizations of the reanalysis performance at daily scale, which might suggest unnecessary changes that could potentially degrade the reanalysis quality. We find that the bias is generally small and varies depending on the canyon and the year. While higher velocities might be expected during years with denser water, such as 2005 and 2012, increasing the modelled velocities could conflict with the model's mass conservation principle, since observations are taken along the canyon axis where the flow speeds are likely higher than the average over the broader area represented by a reanalysis grid point. Moreover, the transport estimates from the reanalysis are already larger than previous estimates, so further increases would likely result in unrealistic transport values. Another aspect is the duration of cascading events, which the reanalysis tends to slightly underestimate overall, except for some overestimation in 2018. Detailed information about where and when the reanalysis performs better is included in the Results section (lines 92-139 of the marked manuscript).

**Minor remarks:**

-L13: As I understood through the manuscript, the monitoring of Cape de Creus started in 2005/2006 for winters and then permanently from 2011, not from 2003 as stated.

You are right, it started in 2005. We have changed this in the 3rd line in the abstract.

-L27: you may want to include the common names for these dry and cold northerly winds (Mistral and Tramontana).

As suggested, we have now included the common names of the Mistral and Tramuntana for clarity (line 29 of the marked manuscript).

L35: basin floor → bottom

We have replaced it for "bottom of the basin floor" to specify that it is at the basin floor, the deepest part of the basin (deeper than the canyon floor).

-L38-40: Any study under CMIP5 or CMIP6 scenario projections?

This is a very interesting point. As far as we know, there are no studies using the models of CMIP5 and CMIP6 for climate projections of DSWC. Also, CMIP5 and CMIP6 have a very low resolution (1º or about 100km), which makes them difficult to study the shelf area.

-L50: (CMEMS, https://marine.copernicus.eu/)

Instead of the general CMEMS website, we used the reference to the specific dataset of CMEMS, that now it is available in SEANOE with the same doi. We have updated the reference in the reference list and in the citation:

*According to the CORA dataset (Szekely et al., 2024) and the SeaDataNet (https://www.seadatanet.org/) observational database, from which this reanalysis assimilates their salinity and temperature...*

*References*

*(...)*

*Szekely, T., Gourrion, J., Pouliquen, S., Reverdin, G. CORA, Coriolis Ocean Dataset for Reanalysis. SEANOE. https://doi.org/10.17882/46219, 2024*

Instead of *(CMEMS, 2023)*.

-L60: from 1987 to 2021→ the reanalysis provides data until 05/31/2023, but you focus your analysis on 1987-2021.

The reanalysis now covers a longer time period and will incorporate additional years in the future. We should have specified that this analysis is based on the period 1987-2021. We have replaced it for "It provides daily averages from 1987" (deleting

"to 2021") so that it does not get out of date. Then, we have added the phrase: "and this study focuses on the 1987-2021 period".

-L88: significant 'anti-correlation' → 'negative correlation'

Thank you for your correction. We have changed it.

- L95: 'temperature minma at CCC' → and LDZ isn't it?

In many occasions, the day with temperature minima in CCC coincides with the day with temperature minima at LDC, but not in all years (for example, in 1987, the minimum temperature in the LDC was a bit later than in the CCC). Thus, we needed to choose one or the other, and we chose CCC.

-L96: (Sv; 1 Sv = 106 m$^3$/s) → (Sv; 1 Sv = 10$^6$ m$^3$/s). Please revise the use of superscripts throughout the manuscript.

Thank you for your notice. Now it is all fixed.

**Reviewer 2:**

**General comments:**

The authors analyze the ability of a numerical modelling (The Copernicus MED-SEA physics reanalysis 2017-2023) to simulate dense water cascading in the Gulf of Lions during winter. The assessment is carried out by comparing simulations with independent observations in the Cap Creus Canyon (CCC) and the Lacaze-Duthier Canyon (LDC), located to the west of the Gulf of Lion. The results compare temperature and velocity near the bottom (30 m above the sea floor) at a depth of 1000 m along the axis of the canyons. Visual and statistical (Taylor diagram) are invoked to support assessment. This is followed by a brief description of the cascading processes modelled, focusing on the winter of 1999 and the role played by shelf water density during 33 years.

Although such modelling is not fully designed to simulate the cascading (relatively coarse resolution, poor representation of canyon bathymetry, z-coordinates, hydrostatic assumption, etc.), it is reasonably successful in triggering the cascade in line with observations. This is the main and interesting result of the paper.

Modelling appears to confirm a 6-7 years cycle of paired cascading events and suggests an inhibition of such processes during EMT. On this point, the paper does not seem complete to me. The authors should study the role of atmospheric forcing (in particular the buoyancy flux over the Gulf of Lion and the wind forcing) in pre-conditioning and triggering the cascading. This could be an interesting complement to previous cascading simulations (Dufaut et al. 2004, Ulses et al. 2008ab), which have focused solely on specific events, or to Hermann et al. (2008), which have dealt with climatological and more regional scales.

Thank you for your comment. We agree that to investigate the role of atmospheric forcing in pre-conditioning and triggering the cascading is an important point; however, it is not the purpose of our study. Our scope is to provide a multidecadal overview of IDSWC events thanks to the use of long-term mooring data and a reanalysis product, which is a completely novel approach and provides an important advance in ocean science. For that, we carry out a number of careful analyses on the IDSWC events, and how the reanalysis performs. Also, we explore why the renalysis is able to reproduce an IDSWC event (by increased shelf water density over the LIW density threshold, Figure 5), and how the modelled dense shelf water cascades along the canyon, at a daily time scale, for over 30 years. For example, in the Supplementary movie S3, you can appreciate the whole process of dense shelf water cascading along the transect of the studied submarine canyons, from the 1st of January to 30 of May on a daily basis, during all the years with IDSWC

since 1987. We believe this resource will be valuable to researchers studying similar processes and to others who rely on this type of information.

We demonstrate that one of the reanalysis datasets publicly available at the Copernicus Marine Service (CMEMS) may be used as a reliable tool to identify overflows in the deep Mediterranean. With this paper, we encourage this kind of study to be tested for other areas, to reach more communities, since CMEMS freely provides reanalysis data for all ocean regions, with different products.

One would expect more deep interpretation of the modelled cascading free period (1988-1997). Is it related closely to the EMT as suggested - i.e. to the water masses characteristics due to a past and distant event - or to the NAO (or others Mediterranean Oscillation indexes) -i.e. to the current atmospheric forcing-?

Our intention was to suggest possible interpretations for the observed interannual variability, including potential links to atmospheric forcing or climate variability. However, since the triggering factors of IDSWC are not explored in this study, we have removed the corresponding sentences from both the Discussion and Conclusion sections to avoid misinterpretation.

Why the 2019 event is missing in simulation (figure 2). The cascading does not reach 1000m deep in the simulation?

That is right. But as written in the text, we do not consider the spike of 2019 in the observations as an IDSWC event for our analyses, because it only lasted one day.

As written in the paper, it seems suspicious as regard to the residence time in the Gulf of Lion to consider a connection between two consecutive events. Are the apparent '*paired events*' triggered by the meteorological forcing?

This is a very interesting topic. Durrieu de Madron et al (2023) also noticed this 2-year pattern, and therefore we added this citation in the text. However, the reason why IDSWC occurs in two consecutive years is still an ongoing discussion. To determine if this is triggered by meteorological forcing, we would need to go beyond the investigation that we have carried out in this manuscript and investigate deeply the triggering factors for IDSWC. However, this is out of the scope of this paper.

In my opinion, the discussion remains too brief and some results require further interpretation, as numerical modelling makes it possible to study the dynamics (as the authors suggest in their conclusion). Otherwise, the preprint is limited to the assessment of the cascading process in a reanalysis in the Gulf of Lion, its scope becomes less ambitious and the title inappropriate. The current title is too generic. The reader is expecting more a numerical experiment rather than an assessment of

a reanalysis in the western part of the Golf of Lions. Please consider rewording the title.

You are right in noting that this is not a numerical experiment in the classic sense. To avoid confusion, and since this is the first time that a reanalysis product has been successful in capturing the dense shelf water cascading in any part of the world, we revised the title to "Capturing dense shelf water cascading with a high-resolution ocean reanalysis". This new title better emphasizes the main scientific contribution of the paper. It is true that we use a specific reanalysis and a specific area, but we aim to highlight the novelty of the approach and its potential relevance to the wider oceanographic community.

I'm suggesting a major revision, because I think that you need to go beyond simply assessing the modelling of cascading to the west of the Gulf of Lions. The success of this modelling calls for an interpretation that is only very briefly sketched out in this version of the paper. Alternatively, it is possible to focus on the assessment, but this also reduces drastically its relevance.

Thank you for your thoughtful feedback. Our aim with this short paper is to provide a focused contribution on how ocean reanalysis products, and specifically the Copernicus Medsea reanalysis can be used to study DSWC, in agreement with long-term in situ observations. This is not a paper intended to describe and examine all the processes involved in DSWC. As such, we have chosen to keep it as a OS format instead of going for the classical paper. We believe that our findings are remarkable and interesting to a broad community, and it sets an advance for the knowledge and methods in the ocean science community.

**Detailed comments**

Line 70. What means exactly *detrended*? Is there a global temperature trend in numerical modelling and or in situ data?

You are right, "detrended" alone is a bit unclear. "Detrended" refers to removing the trend of the whole time series. In fact, there is a significant increasing trend (Mann-Kendall test at 95% confidence level) in temperature in both canyons at 1000 m of depth, according to reanalysis and observational time series. This step was necessary to finely capture the days with intense dense shelf water cascading in all decades, without omitting them in the latest years, or overconsidering them in the first years of the time series. To make it clear in the text, keeping it concise with few words, we have changed ('detrended" by "after detrending"), so the text reads*: (…) daily mean potential temperature (θ) ≤ 12.6°C (after detrending) (…)* (lines 76-77 of the marked manuscript)

Line 70. The expression 'through the sections in Figure 1a' seems more appropriate than '*along the water column*'. Have you calculated the transport from the bottom to the surface by applying your salinity and density criteria or have you implicitly added a depth criterion?

We agree with your comment, and have made the pertinent changes accordingly. Also, no depth criterion was added, only the mentioned temperature, speed, density, and salinity criteria.

Line 70. The detection of IDSWC from *in situ* data is clear (T<=12.6 and downstream velocities >=0.1m/s at 30m from the floor), but the extraction of the same data in reanalysis need to be clarified. I guess the velocities and the temperatures are extracted from the deeper cell (i.e. at ~15m from the floor) in the modelled canyon axis.

The same temperature and speed criterion is used for reanalysis and observations. We have clarified this in the text (lines 77 of the marked manuscript). In fact, it is in the deepest cells of the reanalysis in both canyons, that are the same as those represented in Figure 1b, which we refer to in the same paragraph.

Line 80-85. Why the 2019 event is missing in simulation (figure 2). The cascading does not reach 1000m deep in the simulation?

That is right. But as we have written in the text, we do not consider the spike of 2019 in the observations as an IDSWC event, since it only lasted one day. We have clarified this in the captions of Figure 2.

Line 90. In guess the correlations have been performed only on the winter (JFM + AM?) periods as quoted at the beginning of the next section (line98).

Actually, the correlations are calculated taking into account the whole period shown in Figure 2, including all seasons. And in the next paragraph, it is truly only in winter, from January until May (JFM + AM), as shown in the plots of Figure 3.

Figure 1. The figure 1b is a bit confusing and may be not useful (it is obviously not the model bathymetry and the colored boxes look like horizontal section while vertical cross-sections are defined in figure 1a).

We agree with your comment. It was not clearly described in the text nor in the figure captions. Figure 1b actually represents the real bathymetry (showing the incising canyons), which serves to compare it with the resolution of the reanalysis that is very relevant for the performance of the reanalysis to reproduce the overflows in submarine canyons. This also gives an overview of why the bottom depth of the reanalysis does not match the bottom depth at the canyon axis. This figure helps to

understand the resolution of the reanalysis in comparison with the size of the canyons, and this is the reason why this reanalysis performs that well. To avoid confusion, we have added some explanation in the caption of Figure 1b: *(b) Scheme of the canyons* *with real bathymetry at 200m resolution from EMODnet (https://emodnet.ec.europa.eu/)* *showing the mooring observation sites (white cones) and reanalysis* *grid points* *(colored boxes),* *comparing the resolution of the reanalysis with the width of the canyons**. For clarity, the vertical axis is exaggerated by a factor of 5.*

Line 106-107. Neither figure 3i nor figure 2 show an overestimation of the duration of cascading. You should add the plots for winter 2018 to figures 3a-h to support your affirmation.

According to Figure 2, the 1000 m deep cascading event in 2018 only occurred in the Cap de Creus canyon and not in the Lacaze-Duthiers Canyon. We did not include the plots for winter 2018 in Figure 3a-h because of spatial constraints, so we chose to only include the clearest ones for both canyons (in 2018, the LDC did not have IDSWC). Nonetheless, although it is not in the manuscript, we do have plotted the reanalysis and observations in the Cap de Creus Canyon for winter 2018, and the the simulation clearly overestimates the duration of the observations:

[Figure]

We think that by showing the 2018 in the Taylor diagram in Figure 3i is enough to understand that the simulation overestimates this event, because the standard deviation of both temperature and velocity is larger than that of the observations. We referred to Figure 2 in the text because in the Cap de Creus Canyon (Figure 2c-d), you can see a small spike in the observations (red line, temperature drop and speed increase) before the larger spike of the reanalysis (black line), whereas in the rest of winters, the "red spike" of the observations tends to be larger than the "black spike" of the reanalysis.

Line159-160. The expression '*balance between'* (i.e. equilibrium) does not seem appropriate to me. You probably mean 'contribution of' or …

We agree with your comment, and have replaced it as "contribution of", as suggested.

Line 231. The exact reference seems to be slightly different: https://egusphere.copernicus.org/preprints/2025/egusphere-2025-1310/

Yes, thank you for your notice. The title was slightly modified, and the manuscript was already submitted, and it is currently under review. The reference is now:

Arjona-Camas, M., Durrieu de Madron, X., Bourrin, F., Fos, H., Sanchez-Vidal, A., and Amblas, D.: Dense shelf-water and associated sediment transport in the Cap de Creus Canyon and adjacent shelf under mild winter regimes: insights from the 2021–2022 winter, EGUsphere [preprint], https://doi.org/10.5194/egusphere-2025-1310, 2025.

**Typos**

Line 89. Probably "r=-0.40" instead of "r=0.40"

We agree with this comment and have made the correction accordingly.

Line 95 in figure caption -> $1sv = 10^6 m^3 s^{-1}$

Changed. We have now fixed superscripts where needed.